# Discriminatory Brain Processes of Native and Foreign Language in Children with and without Reading Difficulties

**DOI:** 10.3390/brainsci13010076

**Published:** 2022-12-30

**Authors:** Najla Azaiez, Otto Loberg, Kaisa Lohvansuu, Sari Ylinen, Jarmo A. Hämäläinen, Paavo H. T. Leppänen

**Affiliations:** 1Department of Psychology, University of Jyväskylä, 40014 Jyväskylä, Finland; 2Department of Psychology, Faculty of Science and Technology, Bournemouth University, Fern Barrow, Poole BH12 5BB, UK; 3Department of Teacher Education, University of Jyväskylä, 40014 Jyväskylä, Finland; 4Cognitive Brain Research Unit, Department of Psychology and Logopedics, Faculty of Medicine, University of Helsinki, 00014 Helsinki, Finland; 5Logopedics, Welfare Sciences, Faculty of Social Sciences, Tampere University, 33014 Tampere, Finland; 6Jyväskylä Centre for Interdisciplinary Brain Research, University of Jyväskylä, 40014 Jyväskylä, Finland

**Keywords:** speech perception, native language, foreign language, reading difficulties, MMR, LDN

## Abstract

The association between impaired speech perception and reading difficulty has been well established in native language processing, as can be observed from brain activity. However, there has been scarce investigation of whether this association extends to brain activity during foreign language processing. The relationship between reading skills and neuronal speech representation of foreign language remains unclear. In the present study, we used event-related potentials (ERPs) with high-density EEG to investigate this question. Eleven- to 13-year-old children typically developed (*CTR*) or with reading difficulties (*RD*) were tested via a passive auditory oddball paradigm containing native (Finnish) and foreign (English) speech items. The change-detection-related ERP responses, the mismatch response (MMR), and the late discriminative negativity (LDN) were studied. The cluster-based permutation tests within and between groups were performed. The results showed an apparent language effect. In the *CTR* group, we found an atypical MMR in the foreign language processing and a larger LDN response for speech items containing a diphthong in both languages. In the *RD* group, we found unstable MMR with lower amplitude and a nonsignificant LDN response. A deficit in the LDN response in both languages was found within the *RD* group analysis. Moreover, we observed larger brain responses in the *RD* group and a hemispheric polarity reversal compared to the *CTR* group responses. Our results provide new evidence that language processing differed between the *CTR* and *RD* groups in early and late discriminatory responses and that language processing is linked to reading skills in both native and foreign language contexts.

## 1. Introduction

Dyslexia is a frequent developmental impairment when learning to read and spell; it appears independently of any sensory impairment or other neurological disorder with a prevalence ranging from 5–10 percent [1,2,3,4,5]. Reading difficulties may appear despite an average or above-average level of general cognitive skills and linguistic performance in spoken language and vocabulary [6]. Dyslexia has been linked to problems in developing well-defined phonological representations [7,8] or to problems in accessing them [9,10,11]. These problems have been thought to occur in a large percentage of dyslexic readers [12]. Poor phonological processing skills in dyslexics have been linked to speech perception abilities in a foreign language, including second-language learning [5,13,14]. Despite the phonological processing difficulties behind problems in foreign language learning and its relation to the native language in dyslexia, the background of learning challenges remains poorly understood. Studying brain responses in foreign language processing may make it easier to understand the origin and possible defective mechanisms that may cause learning difficulties in such a context [14,15]. Therefore, the present study aims to a better understand the relationship between native and foreign language processes and to investigate the possible link between language processing and reading difficulties. We are particularly interested in investigating how the discriminatory processing of native and foreign spoken language at the level of brain responses differs between school-aged children with reading difficulties when contrasted to typically reading control children of the same age.

### 1.1. Language Development and Speech Processing in Typical Readers and in Dyslexics

Speech perception is based on mapping basic auditory information into specific phonological units by identifying acoustic features and their boundaries [16]. The perception of foreign speech sounds may rely on the identification of well-established native representations [17,18,19]. However, these representations may also develop independently when the speech sounds of the foreign language do not exist in the native language [20]. Allophonic theory suggests that, in dyslexia, the brain maintains its sensitivity to irrelevant speech contrasts, which may disturb the development of neural networks for categorical speech perception [2,7,14,21,22]. Several studies have shown the link between early auditory and speech perception abilities measured during the first year of life to the later emergence of reading difficulties [23,24,25,26,27]. Researchers from the Jyväskylä Longitudinal Study of Dyslexia [28,29,30] have reported associations between brain activations at six months, preschool-age cognitive skills, and reading development up to 14 years of age [25]. In line with these findings, the longitudinal project of the Dutch Dyslexia Program showed a correlation between early event-related-potential (ERP) responses to speech sounds at the age of two months and later reading skills measured between seven and nine years of age [31,32,33].

### 1.2. Native and Foreign Language Acquisition in Dyslexia

Previous studies on native and foreign language learning have suggested that proficiency in the first language affects second language learning skills highlighting the presence of a possible link between processing the two languages [34,35]. Other studies have suggested possible phonological awareness transfer mechanisms across languages [15,36,37,38]. This question has been addressed mainly from a bilingual point of view [39]. The speech perception of foreign language has been rarely studied in the context of reading difficulties. If difficulties in phonological processing are independent from the language–that is, universal–the processing of both the native language and a foreign language is likely to be deficient in the case of dyslexia. However, it is not fully clear how compromised speech perception skills in the first language may affect the learning of a foreign language [40,41,42]. Learning a foreign language remains challenging for typical readers as some difficulties may emerge when the foreign language contains speech sounds that do not have equivalent representations in the native language. For example, the English sound [ʃ] does not have an equivalent in the Finnish phonology [5]. These difficulties are amplified in the context of reading problems.

Studies with behavioral assessments, mainly in adults, have shown differences between typical readers and individuals with dyslexia in processing of foreign or second language. For example, Soroli and colleagues explored the speech perception of native and foreign languages in adult dyslexics and showed that word stress discrimination was deficient in dyslexic participants in foreign language processing [14]. Several works have supported the view that first-language deficits may affect second-language learning in dyslexics and normal readers [36,41,43,44], but the brain mechanisms involved are not fully understood. Previous findings on the neural level suggest that the brain activations differ in processing native and foreign language, not only in adults but also in typically developed children to [45,46,47,48] (46, p. 150). In a study by Ylinen and colleagues, no atypicalities were found in the brain activations for native language words or in second language speech sound processing in nine- to 11-year-old school children [5]. However, weaker brain activations in the right temporal cortex of dyslexic participants were found in the processing of familiar second-language words. This brain area has been previously reported to play a role in word form learning [49].

The current study adopted a modified version of the two-sequence two-deviant oddball paradigm used by Ylinen and colleagues [5]. The present study investigates the brain responses to native and foreign language processing in 11–13-year-old children with and without reading difficulties. Formal instruction of English (foreign language) started when the Finnish participants of Ylinen et al. and the current study were aged eight and nine. Therefore, we expect this age group to have stronger neural representations for the foreign speech sounds as they have longer exposure to the foreign language compared to Ylinen et al. [5], which was conducted with 9–11-year-old children. The investigation of an older age group may give us a clearer view on foreign language processing in children with reading difficulties and a better understanding of the possible differences in discriminatory brain responses between typical readers and those with reading difficulties. 

### 1.3. Phonological Deficit and Dyslexia in ERP Research

Several studies have shown that speech perception and reading ability are mediated by phonological awareness [12,40,50,51,52,53]. The poor categorization of speech sounds, reflecting deficient internal phonological representations, could be the reason for the developmental deficiency in phoneme identification [54,55,56,57,58]. For example, several behavioral studies have shown that children with dyslexia have poor categorization abilities of consonants [2,59,60].

The ERP components obtained in response to speech stimuli are believed to reflect different processes, from sound detection and feature extraction to categorization [16,61,62,63] (61 p. 89, 62 pp. 14–17). Studies of the neural correlates of auditory and speech perception in dyslexia have applied the auditory oddball paradigm in different versions using different types of stimuli [5,20,25,64,65,66,67]. Some of these studies used, for example, consonant-vowel (CV) syllable stimuli. They were conducted in adults and children and showed atypical mismatch response (MMR) and a late discriminative negativity (LDN) response in individuals with dyslexia [16]. The results showed consistently diminished MMR response in both children and adults with dyslexia when processing difference of tones durations and frequencies. An attenuated MMR response was also reported in children with reading difficulties when processing syllable discrimination. Similar results were also found for the LDN with an attenuated response in dyslexics. These ERPs have been shown to reflect acoustic spectral changes within the spoken syllables in the context of reading difficulties [68,69]. Change detection of the phonological structure investigated MMR and LDN responses using the oddball paradigm with syllable stimuli and was shown to reflect the neural maturational state [70,71]. We analyzed discriminatory ERP responses such as the mismatch response (MMR), commonly labeled in literature as the Mismatch negativity (MMN), and late discriminative negativity (LDN).

### 1.4. The Discriminatory Brain Responses

The MMR component has been largely investigated in auditory [72] and developmental language processing contexts [73], and in relation to reading development [74]. It reflects pre-attentive discriminatory abilities in a pattern regularity violation context [63,75,76]. The MMR is typically elicited in the oddball paradigm and expressed as a negative peak in adults, or as a positive or negative peak in infants and children between 130 ms and 250 ms [77]. The response is visible in the subtraction of the response to a frequently repeated standard stimulus from the response to a deviant stimulus. Moreover, the MMR component has been studied extensively in relation to reading and reading difficulties [65,78,79,80,81] and to foreign language learning [5,20,82].

In addition to the MMR response obtained in the oddball paradigm, several ERP studies have highlighted the presence of a later response at a time frame between 300 ms and 600 ms [83,84,85], which was called the late mismatch response [86,87,88] or, more commonly, termed as late discriminative negativity (LDN) [70,71,78,80,83,89,90]. The LDN response seems to be co-occurring in the MMR-P3a-LDN complex [65,71,84,91,92,93,94], commonly observed in linguistic stimuli, and has been reported to reflect further auditory discriminative and complex cognitive processes [16,65,78,83]. Studying these discriminatory brain responses may give further insight on native and foreign language processing and how it may be linked to reading difficulties.

### 1.5. Hypotheses and Objectives

This study investigates brain responses in typically developing children (*CTR*) and children with reading difficulties (*RD*) while processing native and foreign speech sounds. The paradigm used is based on the two-sequence two-deviant oddball paradigm presented in Ylinen’s study [5]. Our goal is to further investigate whether discriminatory speech processes (MMR and LDN) differ between *CTR* and *RD* groups in native and foreign speech sounds. The participants of this study are two years older than those in Ylinen and colleagues’ study. This age group is expected to have a stronger neural representation of the foreign speech items, so this new data may provide further insights into foreign-language processing. This investigation will also look at the relationship between speech perception in both languages and reading. Assuming a weak quality of the phonetic representations in foreign language compared to the native language phonetic representation in the *RD* group and based on previous studies reporting diminished brain responses in dyslexics when processing native speech sounds [5,65,91,95], we may expect a similar effect on foreign-language processing. Based on previous literature, we may expect both MMR and LDN responses to reflect these weaker neural representations via diminished ERP activations [16,87,92]. Thus, abnormal, reduced, ERP responses are expected in the *RD* group in both languages. These responses are hypothesized to be further diminished in the foreign language processing context compared to the native language processing. However, as we are using the same paradigm and stimuli used in Ylinen et al. [5], we expect to observe possible similarities in the results, although they may be contradictory to previous findings. Ylinen found no group differences between the *CTR* and *RD* groups in processing native words and foreign pseudowords. The amplitude of the MMR response for familiar second-language words correlated with the reading skills in native language [5]. Furthermore, we investigated the time course variation of the response patterns and dynamics occurring at the MMR and LDN time windows in each contrast to investigate the variation of the phonological brain representation within *CTR* and *RD* groups. These were studied to better understand the origin of the group differences and to compare the MMR and LDN responses with previous findings in the literature.

## 2. Materials and Methods

### 2.1. Participants

We report the ERP results of 86 typically reading control participants (*CTR*) and 26 participants with reading difficulties (*RD*) whose data remained valid for analysis after excluding those with poor electroencephalogram (EEG) data quality or an insufficient number of artifact-free EEG epochs (for a detailed accepted number of trials, see the summary Appendix A). The mean age for the control children was 12.36 years (standard deviation (SD) = 0.27; range = 11.78–12.84; 43 females and 43 males), and for the children with reading difficulties it was 12.31 years (SD = 0.34; range = 11.84–12.94; 8 females and 18 males). The participants invited for the EEG recordings were a sub-sample of 440 children from eight different schools in the area of Jyväskylä city in central Finland, who initially participated in the eSeek project (Internet and learning difficulties–A multidisciplinary approach for understanding reading in the new media (eSeek), project number (274022)) [96]. All participants were native Finnish-speaking school children with no history of neurological disorders, head injuries, or hearing problems, based on the parental reports. They were all studying English as a foreign language in school and exposed to the English language daily through media, such as TV channels or the Internet. The groups were sorted based on a reading fluency score derived from three reading tasks (described below) and computed for each participant over the whole sample. The reading fluency score threshold was set below the 10th percentile for the *RD* group and was set at equal to or above the 10th percentile for the *CTR* group. Additionally, all participants had to score above 15 points in the shortened Raven’s progressive matrices test and below 30 points in the Attention and Executive Function Rating Inventory (ATTEX in English, KESKY in Finnish) on the amount of attention and executive function problems. The detailed descriptions for each test are presented below. All participants and their parents signed informed consent forms prior to their participation. The study was conducted according to the Declaration of Helsinki. Approval for the project was given by the ethical committee of the University of Jyväskylä, Finland.

### 2.2. Selection Criteria and Tests

#### 2.2.1. Reading Score

A latent score was computed for reading fluency using principal factor analysis with PROMAX rotation in the IBM SPSS Statistics 24 program (IBM SPSS Statistics for Windows, Version 24.0. IBM Corp; Armonk, NY, USA). This score was estimated with the following three tests: the Word Identification Test, a subtest of the standardized Finnish reading test ALLU [97]; the Word Chain Test [98]; and oral pseudoword text reading [99]. These tests were loaded to the fluency factor as follows: Word Identification Test (0.683), Word Chain Test (0.872), and oral pseudoword text reading (0.653).

The word identification test included 80 items, each consisting of a picture and four alternative written words. The task was to identify and connect the correct picture–word pairs. The score was the number of correctly connected pairs within the two minutes. The word chain test consisted of 25 chains of four words, written without spaces between them. The task was to draw a line at the word boundaries. The score was the number of correctly separated words within the 90 s time limit. The oral pseudoword text-reading test consisted of 38 pseudowords (277 letters). These pseudowords were presented in the form of a short passage, which children were instructed to read aloud as quickly and accurately as possible. The score was the number of correctly read pseudowords divided by the time, in seconds, spent on reading (for a detailed description of these tests, see Kanniainen’s study [100]). The summary of the reading test results for the *CTR* and *RD* groups is presented in Table 1. 

#### 2.2.2. Cognitive Nonverbal Assessment

Participants with a nonverbal reasoning score below the 10th percentile (a score equal to or below 15) in the classroom testing were excluded. This test included a 30-item version of Raven’s progressive matrices test [101]. In this task, partially uncompleted pictures are presented to the child with six different options (six possibilities to complete the pattern), and the child’s task is to identify the correct solution. The performance was timed and the children had a maximum of 15 min to accomplish the task.

#### 2.2.3. The Attention and Executive Function Questionnaire

ATTEX is a questionnaire filled out by teachers [102]. It includes 55 items designed to screen and measure students’ degree of attentional and executive function problems in the school environment. All participants identified with attention deficit (according to their teacher’s rating), and those who scored more than 30 points were excluded from this study, as they exhibited attention problems. The summary of the attention test results for the *CTR* and *RD* groups is presented in Table 1.

### 2.3. Stimuli and Procedure

#### 2.3.1. Stimuli

The auditory stimuli were presented in a passive oddball paradigm, for a total duration of ~20 min. The paradigm was divided into two blocks; the block with Finnish stimuli was presented first, followed by the English stimuli. The stimuli were recorded by a bilingual male native speaker in both Finnish and English and pronounced in a neutral way. These recordings were then screened by native Finnish and English speakers to check for any language bias in pronunciation. The recordings were equalized and normalized in segmental durations, pitch contours, and amplitude envelopes with Praat 5.1.45 [103] and were shortened and resynthesized using the overlap-add method (for a more detailed description of stimuli preparation, see Ylinen’s study [5]. The stimuli consisted of Finnish and English consonant-vowel (CV) syllables that were either words or pseudowords (syllables): *shoe* [ʃʊ:], *shy* [ʃaɪ], and *she* [ʃi:] as the English stimuli and *suu* [sʊ:] (mouth), *sai* [saɪ] (got), and *sii* [si:] (pseudo-word, also a single syllable) as the Finnish stimuli (The spectrograms of the different stimuli are presented in Figure 1).

Finnish phonology does not include the sound [ʃ], so the English items can be easily recognized and identified as a foreign language from the onset of the word [104]. The foreign English stimuli were expected to differ as a function of their frequency of use as words in daily use: *she* is well known and the most frequent of the stimuli, whereas *shoe* and *shy* are known, but less frequent according to the British national corpus [105]. For the Finnish stimuli, *sai* is the most frequent item (the past tense of the verb saada, “to get”), *suu* (“mouth”) is less frequent, and *sii* (a syllable without its own meaning) is the most infrequent according to the Finnish language bank [106]. The Finnish items were chosen as the phonetic equivalents to the English items rather than according to their frequency of use. We prioritized the phonology because it is the most important aspect for across languages comparison. The frequency difference was addressed in the previous paper by Ylinen and colleagues. For a more detailed description of the paradigm and stimuli see [5]. The CV syllable type occurs at a rate of 12.7 percent in the Finnish language (for details, see [107] (p. 65)). 

The stimuli (standard [ʊ:] 800 repetitions, deviant [i:] 100 repetitions, and deviant [aɪ] 100 repetitions) were presented in a pseudorandomized order within each block, with always at least two standard stimuli (and a maximum of five standard stimuli) between the two consecutive deviant stimuli. The inter-stimulus interval between stimuli varied randomly between 850 ms and 1000 ms. The stimuli were presented via a loudspeaker placed on the ceiling approximately 100 cm above the participants’ ear positions and were presented at approximately 65 dB(A). The stimulus volume level was tested with an audiometer before each recording. The sound level meter (Type 2235, Brüel & Kjaer system) was used on a pedestal device at the participant position (settings: sound incidence = frontal; time weighting = fast; ext filter = out; frequency weighting = A; range = 40–110 dB; display = max). Summary of the stimuli properties are available in the Appendix A.

#### 2.3.2. Euclidean Distance and Center of Gravity 

Computation of the Euclidean distance (ED) is commonplace in speech perception and language studies investigating phonological distancing [108,109,110,111,112]. The ED is defined as the scalar sum estimating the difference in phonological/acoustic features between two spoken vowels/items. The ED is applied to a bi-dimensional acoustic space based on tongue position during speech production that correlates with its first (F1) and second (F2) formant frequencies in each item [113]. With the acoustic method, the formant frequencies (F1 and F2) were determined using Praat® software version 6.0.49 [103], and the distance was computed using Excel® 2016 software version 16.0.6742.2048 (Microsoft Corporation. (2022)) by applying the following formula: $$d\left(p,q\right)=\sqrt{\overset{n}{\underset{i=1}{\sum }}{\left(q_{i}-p_{i}\right)}^{2}}.$$

For fricative consonants, computing the center of gravity (COG) is the most common method to calculate the difference in acoustic features between two fricatives. The COG is the phonetic cue in fricative perception consisting of the magnitude weighted average of frequencies present in the fricative spectrum. The COG allows us to distinguish the sibilant fricatives with different places of articulation ([s] vs. [ʃ]) [114]. Importantly, the COG characteristics of a fricative change according to the subsequent vowel (for example, the value for [s] is lower before a rounded vowel, such as [u], than before a non-rounded vowel, such as [i]) [115].

### 2.4. EEG Recording and Pre-Processing

EEG data were recorded in a sound-attenuated and electrically shielded EEG laboratory room located at the University of Jyväskylä. During the measurement, the child was asked to sit comfortably on a chair while watching a muted cartoon movie playing on a computer screen. The child was instructed to minimize his/her movements as much as possible in order to reduce the artifacts in the EEG recording while listening passively to the auditory stimuli. The behavior of the participant was monitored by the experimenters via a video camera. The data were recorded with 128 Ag-AgCl electrodes net (Electrical Geodesic, Inc., Eugene, OR, USA) with Cz as the online reference with NeurOne® software and using a NeurOne amplifier (MegaElectronics Ltd., new designation: Bittium Corporation). The data were sampled online at 1000 Hz, high pass filtered at 0.16 Hz, and low pass filtered at 250 Hz during the recording. Impedances were aimed to be kept below 50 kΩ, and data quality was checked continuously. All necessary adjustments or corrections were performed during short breaks and between the blocks to ensure the best data quality of the recordings.

Brain Electrical Source Analysis (BESA®) Research 6.0 and BESA Research 6.1 (BESA GmbH, Gräfelfing, Germany) were used for offline data pre-processing and averaging. Bad channels that showed noisy data or movements were identified and corrected via signal reconstruction (interpolation) when possible or discarded from the data (number of excluded bad channels: 5.6 [mean]; range: 1–13). Independent component analysis (Infomax applied on a 60 s segment of the EEG; [116]) was used to correct the blinks from each participant’s data. The epochs were set from −100 ms (pre-stimulus baseline) to 850 ms. The artifact detection was set to a maximum threshold of 175 μV for amplitude fluctuations within the total duration of the epoch. A high pass filter of 0.5 Hz, zero phase, was set before the averaging. Bad channels showing noisy data were interpolated using the spherical spline interpolation method [117]. The data were offline re-referenced to the average reference and averaged individually and separately for each stimulus type. Difference waveforms were calculated by subtracting the response to the standard stimulus prior to the deviant stimulus from the deviant response (that is, the deviant minus the standard response). The preprocessing analysis comprised all trials for the deviant stimuli (a total of 100 trials for each deviant stimulus) and the trials before the standard stimulus trials (pre-deviant trials, a total of 200 trials for Finnish standard stimuli and 200 trials for English standard stimuli, 100 for each deviant type). The range, mean number, and SD of the accepted EEG trials in each stimulus type are presented by group and summarized in Appendix A.

### 2.5. Statistical Analyses

Two time windows were used in the current analysis: 150–300 ms was used to investigate the MMR response, and 450–850 ms was used for the LDN response. Statistical differences between the two groups’ brain responses, between the deviant and standard stimuli within each group, and between languages were estimated using BESA Statistics 2.0, with the nonparametric cluster-based permutation tests (BESA, Germany; for the principles of nonparametric cluster-based permutation tests in M/EEG data; see [118]) The number of permutations was set to 1000 for each contrast, and the channel neighboring distance was set to 4.5 cm (with 129 electrodes). False discovery rate (FDR) correction was applied across the permutation tests [119] to correct the *p*-value (FDR correction with *p* = 0.05) performed for the between-language comparisons (Finnish vs. English), between-group comparisons (*CTR* vs. *RD*), and within-group comparisons. The obtained values resulting from the permutation statistics should be viewed as rough estimates; they do not reflect the exact range where the processing differs between the stimuli. To investigate whether the *CTR* and *RD* groups process the Finnish and English stimuli differently, we examined the interaction term in an ANOVA model. A two-factor repeated measures ANOVA was performed (2 [Finnish, English] × 2 [*CTR*, *RD*] mixed ANOVA). The mean voltage was calculated over the full different time windows of each ERP component (same time windows as described above) over the selected electrodes. The selection of electrodes was based on scalp distribution voltage maps and on previous literature as the MMR and LDN responses are typically observed in the fronto-central area [70,91,120]. Eight electrodes were selected: E4, E5, E10, E11, E12, E16, E18, and E19. For a montage illustration, see Appendix A. ANOVA was performed for the difference waves (deviant stimulus–standard stimulus) for both MMR and LDN responses. 

## 3. Results

### 3.1. Native vs. Foreign Language Processing

#### 3.1.1. Comparisons between Native and Foreign Language Processing in the CTR Group

The difference waves (deviant stimulus–standard stimulus) of the English stimuli and their Finnish counterparts in the *CTR* group are presented in Figure 2A,B. The cluster-based permutation test results showed a clear statistically significant enhancement in discriminative responses in the difference wave comparisons for the MMR (~150–300 ms), but not for the LDN (Table 2). The MMR component was enhanced in the foreign contrasts between the *shy*-*shoe* (gray curve) compared with the native counterpart *sai*-*suu* (black curve) and between the *she*-*shoe* (gray curve) compared with the native counterpart *sii*-*suu* (black curve). This discriminative difference was clearly present at the mastoids (see Figure 2B).

#### 3.1.2. Comparisons between Native and Foreign Language Processing in the RD Group

The difference wave comparisons (deviant stimulus–standard stimulus) of the English stimuli and their Finnish counterparts in the *RD* group (Figure 2C,D) showed a statistical difference between native and foreign language in the MMR time window (~150–300 ms) for both comparisons *she*-*shoe* vs. *sii*-*suu* and *shy*-*shoe* vs. *sai*-*suu* (Table 2). The responses to the foreign language were larger than those to the native language. Similarly, for the *CTR* group, no statistical difference was found in the LDN time window.

### 3.2. Group Comparison

The group comparisons of the difference waves (deviant–standard) between the *CTR* vs. *RD* groups in each contrast *sii*-*suu*, *sai*-*suu*, *she*-*shoe*, and *shy*-*shoe* did not show any statistically significant differences between the groups in any of the contrasts (see Appendix A. 

### 3.3. Within-Group Analyses

#### 3.3.1. Within-CTR Group Comparisons for Native Stimuli

The ERP waveforms, amplitude topographies, and cluster-based permutation tests for the control group in both native language conditions, with the deviant stimuli *sii* and *sai*, are illustrated in Figure 3A,B. In the [150–300 ms] time window, the MMR response was observed as enhanced negativity for the deviant stimulus (red curve), with a fronto-central maximum effect between 200–300 ms (~80–180 ms from the CV transition; that is, the onset of the difference between the stimuli), with a slight right-hemispheric preponderance. The MMR response was clearly present at the mastoids with a reversal of the polarity, and less pronounced for the *sai* deviant stimuli than for the *sii* deviant stimuli.

In the statistical cluster-based permutation analyses, the difference between the responses to the deviant stimuli and those to the standard stimuli was statistically significant for both contrasts *sii-suu* and *sai-suu*, with larger responses to the deviant stimuli than those to the standard stimuli. The largest statistical differences appeared between ~180 ms and 240 ms (~60–120 ms from the CV transition) in both native contrasts. The permutation tests showed two clusters: one with a more negative response to the deviant than the standard stimuli at the central area and another with a corresponding more positive response to the deviant at the temporal areas, reflecting the reversal of polarity between the fronto-central and temporal brain areas across the Sylvian fissure. In the late time window [450–850 ms], a slow and late negative fronto-central response (LDN) emerged. The difference was mostly flat, almost null, for the *sii-suu* contrast and more pronounced, with a right-hemispheric preponderance, for the *sai-suu* contrast. In this time window, the difference reached significance after ~550 ms (~430 ms from the CV transition) for *sii-suu*, with only one positive cluster emerging at the occipital area. This effect was somewhat earlier and clearer for the *sai-suu* contrast, emerging at ~500 ms (~380 ms from the CV transition) with two widespread clusters on the frontal and occipital areas. The responses to the deviant stimuli were smaller than those to the standard stimuli. The LDN component was clearly present on the frontal right hemisphere between ~600 ms and 750 ms (~480–630 ms from the CV transition) in the contrast *sai-suu*. This effect was very weak and not clear in the first contrast *sii-suu*.

#### 3.3.2. Within-CTR Group Comparisons for Foreign Stimuli

The ERP waveforms, amplitude topographies, and cluster-based permutation tests for the control group for both deviant stimuli she and *shy* are illustrated in Figure 3C,D, respectively. At the MMR time window, the mismatch effect seemed weaker with a diminished amplitude than that in the native phonologically equivalent conditions. The MMR polarity for the *she-shoe* contrast (Figure 3C) was negative only at the parietal scalp area. The polarity for the *shy-shoe* contrast was clearly positive in the typical MMR fronto-central areas (see difference wave at 150–300 ms, Figure 3D). Cluster-based permutation tests showed a very small negative cluster between ~210 ms and 270 ms in the contrast *she-shoe* and a brief but more robust, larger, and focal frontal cluster at ~210 ms (~90 ms from the CV transition) in the contrast *shy-shoe*. The response to the deviant stimulus *shy* was larger than the standard stimulus response in this time window, in which the effect was almost absent for the stimulus *she*.

The ERP responses in the late time window were similar to those described in the native conditions (see Figure 3A,B for comparison) with the emergence of a typical LDN. In the LDN window, the difference was significant in both foreign contrasts producing two clusters: one negative in the fronto-central area and one positive in the left temporal area. The response was clearer in these foreign contrasts than in the native ones, and it was more pronounced for the second foreign contrast *shy-shoe* than for *she-shoe*, as it was for its phonologically equivalent native contrast *sai-suu* compared to *sii-suu*. The right hemispheric preponderance of the frontal negative cluster was also found for the *shy-shoe* contrast. The cluster-based permutation test results for the *CTR* group are presented in Table 3.

#### 3.3.3. Within-RD Group Comparisons for Native Stimuli

The ERP waveforms, amplitude topographies, and cluster-based permutation tests for the *RD* group in both native language conditions with the deviant stimuli *sii* and *sai* are illustrated in Figure 4A,B, respectively. The *RD* group showed similar ERP patterns to those of the *CTR* group. The ERP pattern of the MMR showed a clear effect on the mastoids in both contrasts. However, the statistical cluster-based permutations did not show significance for the contrast *sii*-*suu* in the MMR time window, unlike the *CTR* group’s response to the same first condition. The difference remained statistically significant with two clusters for the *sai*-*suu* contrast, as it was in the *CTR* group. This central negativity emerged between ~180 ms and 270 ms for this contrast (~60–150 ms from the CV transition). Interestingly, the LDN response did not reveal any statistical differences for the native contrasts in the *RD* group.

#### 3.3.4. Within-RD Group Comparisons for Foreign Stimuli

The ERP waveforms, amplitude topographies, and cluster-based permutation tests for the *RD* group in both foreign language conditions with deviant stimuli *she* and *shy* are illustrated in Figure 4C,D, respectively. The *RD* group showed similar ERP patterns to the *CTR* group, with enhanced amplitude in the foreign language responses. The cluster-based permutation test rendered two clusters for the *she*-*shoe* contrast, with a clear negative cluster on the left frontal area and a widespread positive cluster at the right temporal and occipital areas. In the MMR time window (150–300 ms), significant differences were also found for both contrasts, in which the responses to the standard were larger than those to the deviants. For the second foreign contrast *shy*-*shoe*, a positive cluster emerged at an early time point, ~120 ms, on the frontal left hemisphere, merging gradually to the central area and becoming similar to the *CTR* group cluster pattern observed at this latency (~210 ms). Interestingly, the LDN was not significant for either foreign condition in this group. The cluster-based permutation test results for the *RD* group are presented in Table 4.

The polarity of each cluster indicates the direction of the comparison whether negative (if the second ERP response is bigger than the first) or positive (if the first ERP response is bigger than the second).

### 3.4. ANOVA Results

To test the language × group interaction effect, we conducted 2 × 2 ANOVA separately for the two time windows. The ANOVA results did not show group × language interaction in any of the time windows. Only a language main effect was found, confirming the earlier findings in the between-language cluster-based permutation tests. The ANOVA at the MMR time window yielded a significance for the difference wave comparison, showing a language main effect in the difference between *sai-suu* vs. *shy-shoe*. The ANOVA at the LDN time window also showed a significant effect for the difference *sii-suu* vs. *she-shoe*. In both these difference contrasts, the effect was caused by the larger ERP responses to the foreign stimuli (see the ANOVA results in Appendix A).

## 4. Discussion

We examined the differences of discriminatory ERP brain responses to native and foreign speech sounds in typically developed children (*CTR* group) and in children with reading difficulties (*RD* group). To this end, we used an auditory oddball paradigm and contrasted the discriminatory brain responses between and within languages (Finnish and English) and groups (*CTR* vs. *RD*). The results showed that only MMR, but not LDN, differed between Finnish and English stimuli within both groups. Neither MMR nor LDN showed significant differences when comparing the contrasts between groups. In the within-group analysis, both groups showed a negative MMR with a lower amplitude to native stimuli and a positive MMR to foreign stimuli. The *CTR* group showed significant MMR and LDN responses for all contrasts; however, the MMR response was diminished in foreign language processing and the LDN was somehow weaker in the *sii*-*suu* and *she*-*shoe* conditions than in *sai*-*suu* and *shy*-*shoe*. On the other hand, the *RD* group did not show a significant LDN response in any of the contrasts. The topo maps of the LDN response between 450 and 850 ms showed a reversal of the hemispheric polarity in the frontal area for all the conditions compared to the *CTR* group (see Figure 3 and Figure 4). Overall, the results show a clear MMR difference in language processing in both groups. They also hint at differences in discriminatory brain mechanisms in both typical and poor readers in processing native and foreign speech stimuli. There was also evidence on brain activation variability within language processing, most probably due to the within-stimulus features and semantics. 

### 4.1. Native vs. Foreign Language Processing

To study the role of long-term representations of the native and foreign languages in school-age children, we compared the discriminatory processing of each speech contrast of both languages within the *CTR* and *RD* groups (see Figure 2). The enhanced brain responses in foreign language processing were observed for the foreign-language difference waveforms within the MMR time window in both groups. These waveforms showed that the brain treated the native contrasts in a typical way, as this discriminatory component is typically a negative response. On the other hand, the foreign contrast showed positive polarity over fronto-central electrodes and overall larger amplitude, reflecting enhanced activity. Interestingly, the late processing reflected in the LDN response did not differ between the two languages.

These results showed that the brain responses to foreign language items were different than the native ones within both groups as indicated by the MMR findings. This difference may reflect the instability of the cortical representations in foreign (English) language speech sounds processing, which remain relatively novel items compared with the native sounds even after a long exposure.

Earlier, we hypothesized that less exposure to foreign sounds may lead to weaker representations caused by unstable networks to reflect unfinished or unestablished neural language representation [20,121]. These unstable networks may require more neural resources (either larger cortical area activation or multiple sources) to process the sounds, leading to large, positive responses. For example, processing a foreign sound may recruit additional brain processes, such as a higher activation of the early auditory arousal-attentional mechanism during the early 100 ms post stimulus (P1/N1 response; [122,123]), which may overlap with the first discriminatory response, the MMR. Another possible interpretation of this result is that the enhancement observed in foreign-language processing could be due a specific neural response to differences in the physical features within the stimuli, notably the effect of the early foreign sound [ʃ]. It is also possible that a number of these explanations may co-occur.

In the language comparison, a significant difference was observed in the MMR but not for the LDN response within *CTR* and *RD* groups. This result was supported by the ANOVA findings as it indicated a language effect.

### 4.2. Group Comparison CTR vs. RD

Our results did not show any statistically significant differences between the groups in any of the components. In native language context, a similar result was previously reported by Ylinen and colleagues, as no significant group difference was found in native language word form processing [5]. However, the authors reported a significantly weaker MMR in the *RD* group when processing a second language familiar word (the word *she*). This difference between our results and the previous study’s results in this contrast could be explained by a possible attenuation of this response in our group sample due to a longer exposure to the foreign language and to the age difference of the participants between the two studies. As Ylinen et al. originally reported “a weak MMR”, this response may have further diminished with age until it disappeared. Thus, foreign-language processing, and particularly this contrast, may have reached a mature native-like language processing after some years of exposure. Ylinen et al. did not investigate the LDN response; the authors only focused on the mismatch response. Our results did not show group differences in this component despite the absence of the LDN significance within all conditions in the *RD* group (see Table 4). As an example of a previous study investigating both MMR and LDN responses to speech sounds (vowels discrimination task), Froyen and colleagues reported no discrimination problem of the speech stimuli in the dyslexic group who showed similar brain responses to the *CTR* group [124]. Our results show a similar result, as we did not find any statistically significant differences in any of the components in the group comparison. Some previous studies reported group differences in the MMR responses between *CTR* and *RD* groups, but those studies used different stimuli and paradigms as they investigated nonlinguistic stimuli such as tones [90,125] or synthetic speech [126], which make it difficult to compare with our findings.

When it comes to the foreign-language processing, our study is the first of its kind to investigate both discriminatory brain responses MMR and LDN, in foreign-language processing in the context of good and poor reading. To our best knowledge, only one study has investigated the MMR in dyslexics versus controls in foreign language context [5]. On the other hand, there is no previous evidence in the literature about the LDN response in dyslexics when processing a foreign language. When comparing the two groups, we were able to observe an overall tendency to larger responses in the *RD* group compared to the *CTR* group responses (see Appendix A). This enhancement may indicate extra neuronal activations in the *RD* group when processing the different stimuli. Enhanced ERP responses have also been described in other studies to reflect the less efficient linguistic performance in discriminatory processes [73,80,127]. This is explained as a greater processing effort [73], which may lead to the activation of a compensatory mechanism when processing speech stimuli [80]. In the literature, neuronal networks have been described as remaining open to relevant and irrelevant speech sounds in case of dyslexia as described by the allophonic theory [22]. This higher sensitivity may play a role in compensating for the phonological deficit, expressed in our results as larger amplitude and reflected in higher neuronal activity.

In native and in foreign language processing, neither MMR nor LDN showed a statistically significant difference between the *CTR* and *RD* groups, despite a hint on possible processing deficits in the *RD* group as indicated by the larger brain response.

### 4.3. Within Group Results

#### 4.3.1. Native and Foreign Language Processes in Typical Readers

Brain responses to standard native (Finnish) speech stimuli in the *CTR* group revealed overall typical brain dynamics like those reported earlier in the literature [128,129]. The statistical comparison revealed a difference in processing the deviant and standard stimuli showing the presence of a typical MMR and LDN responses, except for the first contrast *sii*-*suu* showing a less clear response with only a positive cluster over the occipital area.

In the foreign contrasts, the statistical comparison revealed a difference in processing the deviant and standard stimuli, showing the presence of an MMR and LDN responses equally. However, the MMR was very weak and atypical compared to that obtained with the native language stimuli. It showed a very small negative cluster over the central area for the condition *she*-*shoe*, and a positive and very brief cluster for the *shy*-*shoe* condition (see Figure 3C,D). This result may indicate an early effect related to the presence of the foreign onset sound [ʃ] (pronounced *sh*) at the beginning of the English stimuli. 

In addition, atypical responses could be attributed to the coarticulation effect. This sound may produce a different transition cut between the consonant and vowel compared to the initial [s] natively present in the language producing a different brain activation pattern. Formant transitions are important perceptual cues in speech processing. Their shapes vary according to the neighbor consonant affecting the identification of the following vowel [130]. This foreign sound may generate extra attentional processes that may overlap with the MMR response in the foreign language context. Moreover, the response to the fricative [ʃ] could partially encode information coming from the vowel and diphthong that represents a combined information fricative-vowel simultaneously. Generally, in the case of a strong vowel, the information could be present on the prior consonant [131,132] or, in our case, the pre-vowel fricative, producing a specific and distinctive articulatory configuration for each of the initial consonants ([s] and [ʃ]) in each of the stimuli [133,134].

##### The Mismatch Response within the CTR Group

We observed larger negativity in response to the deviant stimuli than in response to the standard stimuli in native language processing, suggesting that typical MMR was generated as a marker of change detection [72,135,136]. The MMR in our study clearly diminished for both foreign language contrasts and created a topographic pattern tending toward positivity. The positive MMR is thought to reflect a mismatch response, which is usually reported in infants and less frequently in older children [67,85]. A clear difference in MMR response between native and foreign stimuli could be seen in our results. However, Ylinen and colleagues reported a negative mismatch response to the same foreign stimuli in younger children (around nine years old) [5]. Thus, it is more likely that the positive MMR observed in our results came from the larger overlapping of an attentional response to the foreign contrasts, an early P3a overlapping with the MMR. The different analysis and filtering settings used in our study compared with those in Ylinen’s work may also be the origin of different results.

Although these remain valid possibilities, our results clearly show that foreign and native stimuli generated different brain responses, with a clear MMR response to the native stimuli, that was less typical in foreign stimuli context reflecting different brain responses in discriminatory processes between native and foreign languages.

These MMR results may also hold the acoustic distance effect between speech items and between different languages, as the MMR has been previously shown to be sensitive to acoustic distance [137]. Based on previous findings, the perceptual stimulus dissimilarity could be quantified via the Euclidian distance, where the ED between standard and deviant stimuli may partially explain the size of the discriminative responses. The ED analysis (see Appendix A) showed that English and Finnish vowels were acoustically distinct and that the *ii* ([i:]) deviant was acoustically further in ED from the standard *uu* ([ʊ:]) than the deviant *ai* ([aɪ]). In the *CTR* group, the responses to the Finnish language may be explained by the ED. The larger the ED between the standard and deviant stimuli, the more pronounced the MMR. However, this effect was not reproduced with the foreign stimuli. Hypothesizing that the brain representations are weaker for the foreign language, we would assume that the brain would rely more on the acoustic features; however, our results suggest that the ED did not play the major role in the ERP responses to foreign stimuli. We think that the early identification of a foreign sound [ʃ] may initiate a different process, indicating clearly that the brain did not rely only on the acoustic properties but other processes may have contributed to the response. Similar conclusions have been reported earlier in Ylinen’s study [5]. Previous studies in control participants exploring sub-phonemic vowel contrast perceptions (the difference between equivalent speech items) showed the sensitivity of the MMR component to the phonetic distance between the stimuli [137]. This highlights the specificity of the brain processes recruited for each language processing.

##### The Late Discriminative Response within the CTR Group

The late discriminative processing (LDN, 450–850 ms) for all native and foreign stimuli was more negative for the deviant than for the standard stimuli; however, the LDN response was not significant in any of the contrasts. Our results showed a frontal negativity with a preponderance to the right hemisphere in response to *sii*-*suu* and *sai*-*suu* contrasts, more left centrally oriented in the *she*-*shoe* contrast and right centrally oriented in the *shy*-*shoe* contrast. The LDN response to the native *sai*-*suu* contrast was more prominent than that to the *sii*-*suu* contrast in the native language processing. The response to the *shy*-*shoe* contrast was also larger than that to the *she*-*shoe* contrast.

Different interpretations of the LDN role have been proposed in the literature, but the functional significance of this component remains speculative because no clear evidence is available in the literature about its exact role [138,139]. This late negativity over the fronto-central area is known to have been generated in response to complex auditory stimuli, such as linguistic stimuli [83]. A recent study showed that LDN is a marker for phonological complexity [70]. The LDN response was previously studied as an index of foreign phonological contrast discrimination [93] and as an indicator of speech perception development [127]. The LDN response may also indicate a coarticulation effect contributing to the late response by generating a stronger response for deviant stimuli carrying the diphthong *ai* (in *sai* and *shy*) compared with the vowel *ii*. This may explain the stronger response obtained with both native and foreign stimuli carrying the diphthong *ai*, which possibly needed an additional neural activation.

Previous studies of the LDN response have shown that this response is not only linked to the complexity of the stimuli [140], but also comprises high-order cognitive processes [89]. We think that, in addition to the coarticulation present in the features (vowel vs. diphthong), additional processes may be involved in the late response, and they are linked to the functions and semantics of the words [84]; *sii* as a pseudoword would produce smaller activation than *sai*, which comprises a function and a meaning (verb = got, past tense). *She* as a word would also produce a smaller activation as a familiar English pronoun than *shy*, which is less familiar and has a complex function (adjective). The difference in processing words vs. pseudowords was earlier highlighted in the early discriminatory response MMR, where distinct responses were reported [141]. The enhancement was interpreted as an indicator of the long-term memory traces for spoken words, which make it one the most plausible explanations. The modulatory effect of the lexical meaning on the brain discriminatory response was earlier shown to offer a processing advantage for the meaningful items [142]. However, all these interpretations remain hypothetically possible.

#### 4.3.2. Native and Foreign Language Processes in Poor Readers

##### The Mismatch Response within the RD Group

The *RD* group showed statistically significant difference for the *sai*-*suu*, *she*-*shoe*, and *shy*-*shoe* contrasts between 150 and 300 ms, but not for the *sii*-*suu* contrast. Interestingly, this same contrast showed a very weak MMR response in the *CTR* group. The more asymmetric activation in this contrast may indicate atypical brain responses in the *RD* group. The weaker and atypical MMR activation in this contrast compared to responses to the other contrasts, maybe due to the nature of the deviant stimulus as a non-word engaging a different encoding strategy compared to real words. Our results also suggested that the response in the *RD* group was different than that to the *CTR* group, which may reflect a different processes in encoding strategy of the same non-word. The positive MMR that was reported earlier in typical readers when processing a foreign language was reproduced in the *RD* group in the same contrasts. The electrical distribution maps within this group showed atypical activations, mostly in foreign language processing. Atypical brain activity in response to speech sound contrasts is supportive evidence of the phonological system deficit that has been previously described in the literature [64]. Smaller MMR amplitude responses to syllables were earlier reported in the *RD* group compared with the *CTR* group [88], and later investigation showed that this effect was speech-specific [143].

The nativeness of the stimuli did not seem to play a major role in the early processing phase in the *RD* group, suggesting weaker and less sensitivity to the initial sound. This effect could be explained by a memory encoding deficit in the *RD* group, as previously reported [144,145]. Phonological deficit and memory impairment were shown to share neuronal mechanisms in dyslexic children between 10 and 14 years old [146]. Thus, a deficit in memory construction with a phonological deficit may be a valid explanation for the results found in the MMR time window within the *RD* group.

##### The Late Discriminative Response within the RD Group

The late discriminative component (LDN) between ~450 ms and 850 ms did not show significant differences between the deviant and standard stimuli responses in any of the contrasts within the *RD* group. Commonly, in the auditory paradigms investigating discriminative processes, the later negative response was reported as part of the complex MMR-P3a-LDN [65,91,93]. The functionality of this response remains largely debated. It is thought to be related primarily to discriminative processing, but further complex processes are also considered to take part in the LDN response [83], such as sound structure processing [70], and attentional processes [94,147].

The absence of any LDN significant effect may suggest reduced abilities of auditory discriminative processing in this group compared to the *CTR* group [16,83], as has also been pointed out in the MMR findings. Weaker LDN responses in dyslexics are expected because they have been reported in the literature [92]. The LDN response was also suggested to reflect neuronal phonological representations [127] and was shown to be modulated by the phonological complexity present in linguistic stimuli [70].

Earlier, we hypothesized weaker neuronal representations in the *RD* group, and our results may indicate such an effect. Weaker representations may produce lower amplitude responses. The complexity (in the case of a diphthong), the coarticulation effect, and the nativeness of the stimuli (Finnish vs. English), as discussed above, did not seem to affect this response because the LDN effect was absent in all the contrasts.

### 4.4. Strengths and Limitations

The complexity of the stimuli maybe considered as limitation since it makes it harder to interpret the results. It is however also a strength of this study because we used natural speech with two deviants in two different languages. Natural speech stimuli in this type of design were rarely investigated in previous studies of dyslexia.

Furthermore, the larger amplitude variation is typically found at the end of the ERP epoch. The lower number of participants in the *RD* group may have increased the signal-to-noise ratio compared with the *CTR* group, which may have caused the results to not reach significance during this time window. Moreover, weak responses combined with the high variability within the group and noisier responses among the participants may induce the suppression of the LDN effect. This smaller *RD* group size compared to the *CTR* group size maybe considered as a limitation in the current study as is causes lower statistical power for some ERP effects.

Another possible limitation in the current analysis is the frequency of use of some speech items, which may have had some effects on the discriminatory responses. However, it is unlikely that this was the case based on the previous study by Jacobsen and colleagues who showed no effect of word frequency/familiarity on the MMR response [142]. 

Although the current study does not directly investigate the relationship between reading scores and the brain responses via a direct correlation analysis, the group definition and the analysis were based on those reading scores. The direct correlation analyses between the brain responses to speech items and the reading scores were previously conducted using the same group’s data and results showing the direct link between reading scores and speech processing are available in our previously published research [148].

## 5. Conclusions

Our results showed that both *CTR* and *RD* groups discriminatory process, as indexed with the MMR response, were different for the native compared to foreign language. They provide new evidence on foreign speech processing, both in typical readers and in children with reading difficulties. Furthermore, our results showed effects of the within-stimulus features and semantics as they seem to affect the ERP responses in both groups. Further investigation is needed to examine in depth the origins of these differences between typical readers and children with reading problems. Our study is, to our best knowledge, the first brain-based evidence on the late discriminative processing in foreign language context and in the context of reading difficulties.

## Figures and Tables

**Figure 1 brainsci-13-00076-f001:**
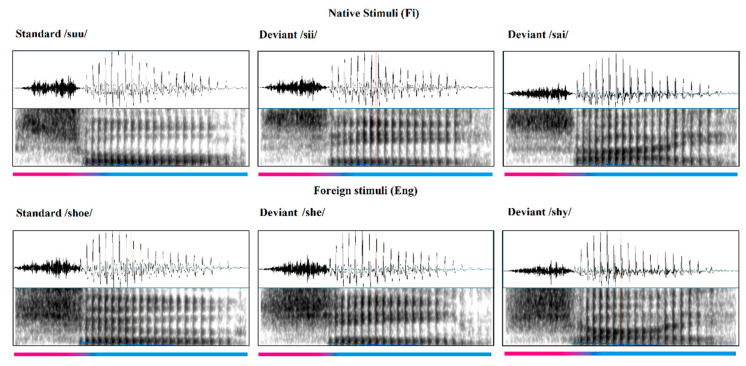
Spectrograms of the stimuli used in the oddball paradigm. At the top are the native (Finnish) stimuli and at the bottom are the foreign (English) stimuli. The color bar below shows the sound change from the fricative (in pink) to vowel(s) (in blue) at around 120 ms.

**Figure 2 brainsci-13-00076-f002:**
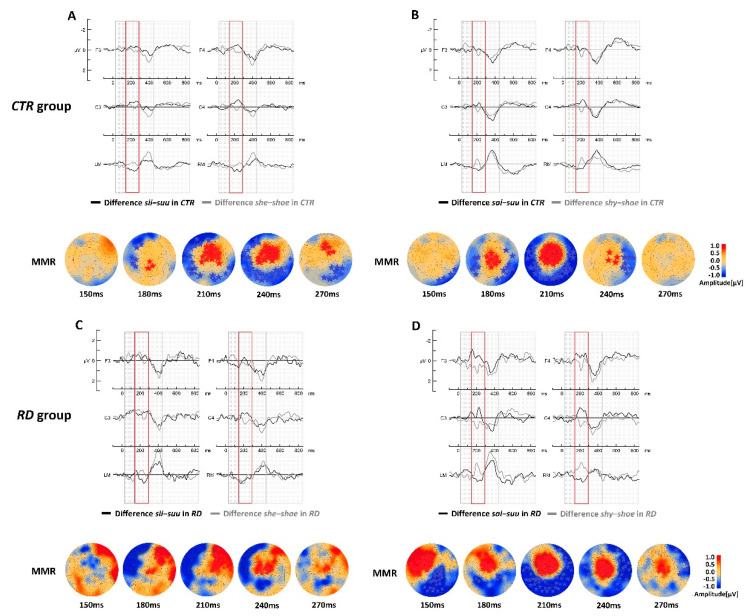
In the top panel, difference waveforms of *CTR* group (**A**) for the contrasts *she*-*shoe* (in gray) vs. *sii*-*suu* (in black) and (**B**) for the contrasts *shy*-*shoe* (in gray) vs. *sai*-*suu* (in black). The statistically significant difference topographies (English–Finnish) in the cluster-based permutation test results are presented below the corresponding waveforms. In the bottom panel, difference waveforms of the *RD* group (**C**) for the contrasts *she*-*shoe* (in gray) vs. *sii*-*suu* (in black) and (**D**) for the contrasts *shy*-*shoe* (in gray) vs. *sai*-*suu* (in black). The statistically significant difference topographies (English–Finnish) in the cluster-based permutation test results are presented below the corresponding waveforms. F = Frontal; C = Central; ML = left mastoid; MR = right mastoid.

**Figure 3 brainsci-13-00076-f003:**
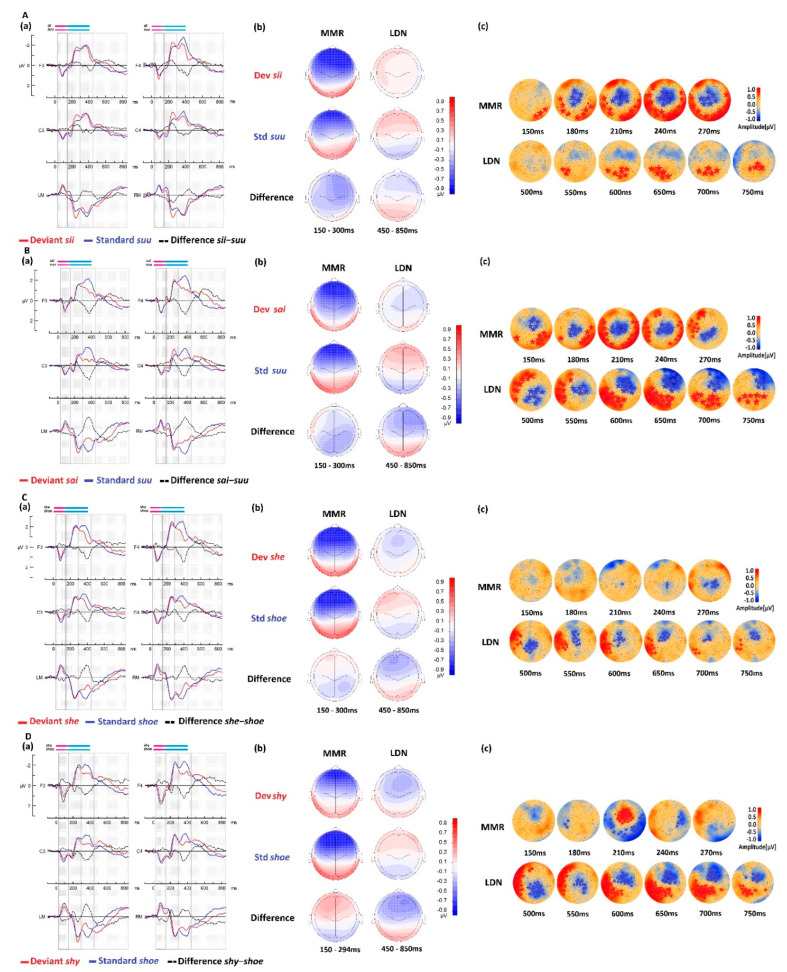
Average brain responses of the control group (*N* = 86) to the native conditions (**A**) *sii*-*suu* and (**B**) *sai*-*suu* and to the foreign conditions (**C**) *she*-*shoe* and (**D**) *shy*-*shoe*. (**a**) ERP waveforms of the native language deviant in red, the standard in blue, and their difference wave deviant-standard in black. The windows of interest showing the MMR and the LDN components are highlighted in black boxes. (**b**) Corresponding means topographic maps over the MMR and LDN time windows. (**c**) The statistical cluster-based permutation test results showing significant differences between the responses to deviant and standard stimuli are indicated with stars. Blue and red colors indicate negative and positive amplitude values, respectively. The measuring unit is µV.

**Figure 4 brainsci-13-00076-f004:**
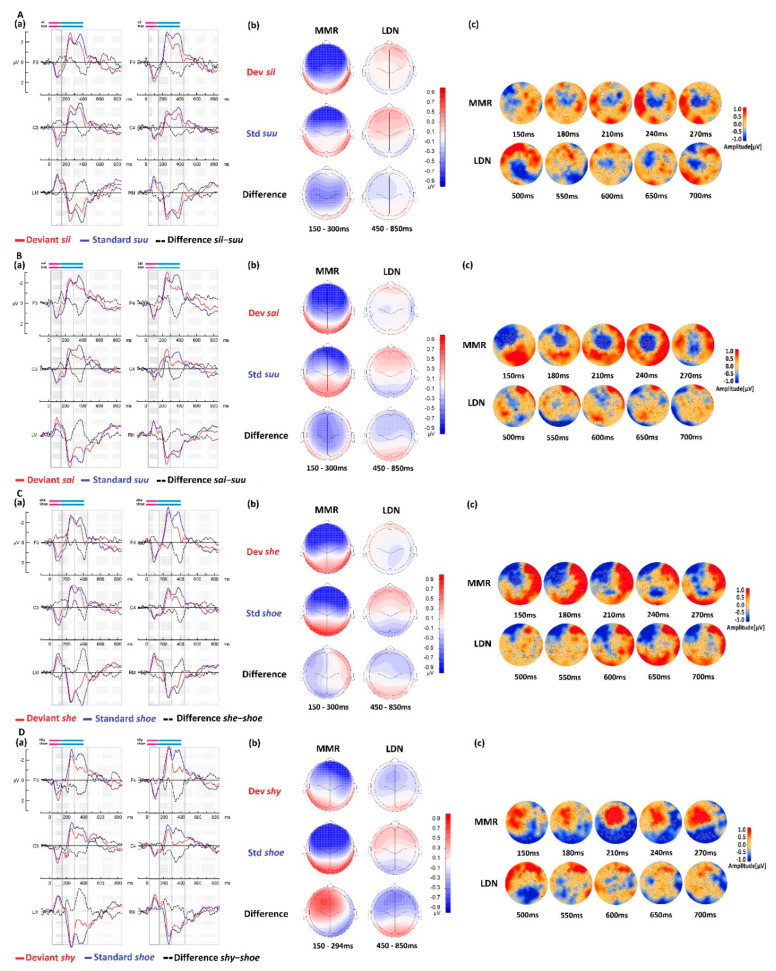
Average brain responses of the reading difficulties group (*N* = 26) to the native conditions (**A**) *sii*-*suu* and (**B**) *sai*-*suu* and to the foreign conditions (**C**) *she*-*shoe* and (**D**) *shy*-*shoe*. (**a**) ERP waveforms of the native language deviant are shown in red, the standard in blue, and their difference wave deviant-standard in black. The windows of interest showing the MMR and the LDN components are highlighted in black boxes. (**b**) Corresponding means topographic maps over the MMR and LDN time windows. (**c**) The statistical cluster-based permutation test results showing significant differences between the responses to deviant and standard stimuli are indicated with stars. Blue and red colors indicate negative and positive amplitude values, respectively. The measuring unit is µV.

**Table 1 brainsci-13-00076-t001:** Summary of the three reading tests (ALLU, Word Chain reading and Pseudoword reading) and the Raven’s Standard Progressive Matrice test for the *CTR* and the *RD* groups.

Group		ALLU			Word Chain			PW Reading			RAVEN		
	df	t-Value	*p*-Value	Cohen’s d	t-Value	*p*-Value	Cohen’s d	t-Value	*p*-Value	Cohen’s d	t-Value	*p*-Value	Cohen’s d
*CTR* vs. *RD*	111		<0.001	1.824		<0.001	2.017		<0.001	1.39		0.192	0.26
			**M**	**SD**		**M**	**SD**		**M**	**SD**		**M**	**SD**
*CTR*		4.36	52.39	8.80	0.94	45.39	12.66	3.23	32.96	3.40	3.14	22.95	3.37
*RD*		1.00	36.69	7.88	0.99	22.23	5.89	1.25	27.11	6.19	3.95	22.11	2.59

Note: *RD* = the group with reading difficulties; *CTR* = the control group; Cohen’s d = the effect size; M = Median, SD = standard deviation of each test in the two groups. The FDR correction alpha value is 0.05.

**Table 2 brainsci-13-00076-t002:** Cluster-based permutation test results of comparison between English and Finnish difference waves for the *CTR* group and for the *RD* group.

	MMR (150–300 ms)	LDN (450–850 ms)
***CTR* group**		
*she-shoe* **vs.** *sii-suu*	Cluster [171–279 ms], Positive (max. 231 ms), *p* < 0.001 Cluster [170–294 ms], Negative (max. 253 ms), *p* < 0.001	ns
*shy-shoe* **vs.** *sai-suu*	Cluster [166–252 ms], Positive (max. 208 ms), *p* < 0.001 Cluster [150–254 ms] Negative (max. 206 ms) *p* < 0.001	ns
***RD* group**		
*she-shoe* **vs.** *sii-suu*	Cluster [152–276 ms], Positive (max. 195 ms), *p* < 0.005	ns
*shy-shoe* **vs.** *sai-suu*	Cluster [150–277 ms], Positive (max. 205 ms), *p* < 0.001 Cluster [150–281 ms], Negative (max. 239 ms), *p* < 0.002	ns

Note. The statistical information in each column represents the cluster range, polarity, time point of maximum amplitude, and *p*-value, respectively. ns = non-significant. All the results were FDR-corrected and only results that survived the FDR corrections were included in the table.

**Table 3 brainsci-13-00076-t003:** Summary of the within *CTR* group statistics using cluster-based permutation tests for the native (Finnish) and foreign (English) conditions.

Condition (Deviant–Standard)		
	MMR (150–300 ms)	LDN (450–850 ms)
*sii-suu*	Cluster [154–300 ms], Negative (max. 246 ms), *p* < 0.001 Cluster [150–300 ms], Positive (max. 258 ms), *p* < 0.001	Cluster [532–800 ms], Positive (max. 661 ms), *p* < 0.02
*sai-suu*	Cluster [150–300 ms], Negative (max. 203 ms), *p* < 0.001 Cluster [150–300 ms], Positive (max. 217 ms), *p* < 0.001	Cluster [450–849 ms], Negative (max. 586 ms), *p* < 0.001 Cluster [450–850 ms], Positive (max. 640 ms), *p* < 0.001
*she-shoe*	Cluster [207–295 ms], Negative (max. 294 ms), *p* <0.02	Cluster [450–849 ms], Negative (max. 503 ms), *p* < 0.005
*shy-shoe*	Cluster [183–234 ms], Negative (max. 209 ms), *p* < 0.001 Cluster [175–275 ms], Positive (max. 214 ms), *p* < 0.001 Cluster [231–295 ms], Positive (max. 257 ms), *p* < 0.03 Cluster [263–297 ms], Negative (max. 290 ms), *p* < 0.03	Cluster [450–850 ms], Positive (max. 575 ms), *p* < 0.001 Cluster [450–849 ms], Negative (max. 671 ms), *p* < 0.001

Note. The statistical information in each column represents the cluster range, polarity, time point of maximum amplitude, and *p*-value, respectively. ns = non-significant. All the results were FDR-corrected and only results that survived the FDR corrections were included in the table. The polarity of each cluster indicates the direction of the comparison whether negative (if the second ERP response is bigger than the first) or positive (if the first ERP response is bigger than the second).

**Table 4 brainsci-13-00076-t004:** Summary of the within *RD* group statistics using cluster-based permutation tests for the native (Finnish) and foreign (English) conditions.

Condition (Deviant–Standard)		
	MMR (150−300 ms)	LDN (450−850 ms)
*sii-suu*	ns	ns
*sai-suu*	Cluster [150–300 ms], Negative (max. 232 ms), *p* < 0.001 Cluster [150–289 ms], Positive (max. 156 ms), *p* < 0.003	ns
*she-shoe*	Cluster [150–300 ms], Positive(max 177 ms), *p* < 0.001 Cluster [150–259 ms], Negative(max. 199 m), *p* < 0.005	ns
*shy-shoe*	Cluster [150–295 ms], Negative (max. 209 ms), *p* < 0.001 Cluster [182–294 ms], Positive (max.294 ms), *p* < 0.002	ns

Note. The statistical information in each column represents the cluster range, polarity, time point of maximum amplitude, and *p*-value, respectively. ns = non-significant. All the results were FDR-corrected and only results that survived the FDR corrections were included in the table.

## Data Availability

The data presented in this study are available in Figure 2, Figure 3 and Figure 4 and Appendix A. Original data can be made available by the authors upon reasonable request.

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
