# Peer review of "Discriminatory Brain Processes of Native and Foreign Language in Children with and without Reading Difficulties"

_brainsci, 2022, doi:10.3390/brainsci13010076_

Round 1
Reviewer 1 Report
My background is behavioral speech and language research and data analysis. I have limited my comments to the scientific justification, presentation, description of analytical methods, and support for conclusions. I cannot comment on the EEG methodology of this work or the precise interpretation of the EEG results, as my experience analyzing EEG data is limited.
The authors make a compelling case for examining native and foreign language processing in their introduction. The authors should edit the first paragraph to remove any references to "possible interactions between language processing and reading difficulties" as they never link EEG responses to behavioral performance.
The EEG methods appear to be sound; although a peer review from an expert in EEG methodology should be sought.
The authors need to clarify the justification for their stimuli selection. As they bring up in the discussion the fricative /sh/ sound starting all of the English words provides an early indicator for a foreign word. Please provide a rationale for having the foreign stimuli begin with an obvious distinction; esp when there are English stimuli (e.g., sue, see, sigh) that would have not included the fricative /sh/ while maintaining the vowel of interest. Further, the authors describe selecting CVV stimuli; however, they in fact selected CV stimuli. The Finnish stimuli are SPELLED with two vowel graphemes, but are produced with a single vowel phoneme, whereas the English stimuli would be considered CCV if going by spelling and not production. Please correct this information (stimuli are CV) in the text. The link between phonemes and graphemes is a key part of reading and the driver behind why phonological awareness is important for alphabetic languages - esp English.
No problems with the analytical description or methods.
My main comment for the results is the appropriateness of discussing location (i.e., mastoids) for EEG. My only experience with EEG and topo maps was to not discuss location because signals from a different part of the brain can be picked up by an electrode "far" away. Again an expert in EEG would be able to provide more detailed feedback on this tendency.
I appreciate the detailed discussion about how the format information and co-articulation of /sh/ may have affected brain responses.
Overall, the authors do a good job of placing their results within the context of the broader literature. However, the discussion would still benefit from editing to improve clarity and readability. The clarity of the discussion could be improved with more informative subheaders, key points placed at the end of paragraphs, and more paragraph breaks to space the text out.
There is no discussion of the strengths/limitations of the work in the discussion. Eg. the small (n = 26) sample size of children with RD. Not being able to account for differences in English exposure beyond school (do some children get more practice than others). I encourage the authors to address at a minimum the small sample size limitation and how their work can still more us forward.
The introduction and discussion could be improved by mirroring the topics in both sections. Section headers in the introduction would help readers glean the main themes more easily (speech perception and reading; native vs foreign processing; EEG work on speech perception). I think the addition of a section on "complications with stimuli in speech perception" in the introduction will better prime the reader for the authors discussion on /sh/, sub-phonemic information, and lexical information. The material is all there in the introduction; it just needs to be organized and tightened up a bit more.
Beyond EEG responses to stimuli and some group differences, I am not sure how these data point to how speech perception skills affect measureable behavior. Please either discuss this limitation in the discussion or perform some sort of that links EEG responses to behavior. Perhaps collapsing across the groups (because the ANOVA showed no group differences) and performing a regression using amplitude (or extractable statistic) to predict reading scores.
Author Response
Responses to review report 1
Review comment 1:
My background is behavioral speech and language research and data analysis. I have limited my comments to the scientific justification, presentation, description of analytical methods, and support for conclusions. I cannot comment on the EEG methodology of this work or the precise interpretation of the EEG results, as my experience analyzing EEG data is limited.
The authors make a compelling case for examining native and foreign language processing in their introduction. The authors should edit the first paragraph to remove any references to "possible interactions between language processing and reading difficulties" as they never link EEG responses to behavioral performance.
Answer to comment 1:
The reviewer raised an important question. Although this paper does not look directly at the correlations between brain responses and reading skills, the whole analysis was based on the group division (CTR vs RD), and that grouping was based on behavioral evaluation (please see the grouping criteria detailed in the method section L215-219). Furthermore, we earlier investigated in our previously published paper the direct correlations between the reading scores and the brain responses to speech stimuli and to visual brain responses in reading. Our previous study investigated the brain responses in speech processing and the reading scores, via correlation analyses, in the same groups/subjects examined in the current manuscript. As evidence for this direct correlation between reading skills and brain activations of speech processing with the same data, we added the reference to that study at the end of the discussion section (L808-818).
Reviewer comment 2:
The EEG methods appear to be sound; although a peer review from an expert in EEG methodology should be sought.
Reviewer comment 3:
The authors need to clarify the justification for their stimuli selection. As they bring up in the discussion the fricative /sh/ sound starting all of the English words provides an early indicator for a foreign word. Please provide a rationale for having the foreign stimuli begin with an obvious distinction; esp when there are English stimuli (e.g., sue, see, sigh) that would have not included the fricative /sh/ while maintaining the vowel of interest.
Further, the authors describe selecting CVV stimuli; however, they in fact selected CV stimuli. The Finnish stimuli are SPELLED with two vowel graphemes but are produced with a single vowel phoneme, whereas the English stimuli would be considered CCV if going by spelling and not production. Please correct this information (stimuli are CV) in the text. The link between phonemes and graphemes is a key part of reading and the driver behind why phonological awareness is important for alphabetic languages - esp English.
Answer to comment 3:
To clarify this matter, the rationale for the use of fricative /sh/ is that we consider it important that the language of the stimuli is evident and unambiguous right from the beginning of each stimulus. This is because the recognition of the language affects the recognition of the words. The results would have been difficult to interpret if the interpretation of the words varied across the sequence (e.g., recognizing stimuli as English at a later time point at the beginning of the sequence than at the end of the sequence) or the time point of the recognition varied between stimuli or between individuals. The stimuli were the same as in the previous study Ylinen et al. (2019), where the frequency of use was addressed: ‘She’ is very frequently used and certainly familiar to the children, whereas ‘shy’ is less frequently used. This was explained in the manuscript.
We thank the reviewer for pointing this out. We agree with the reviewer that it is better to use CV to describe the English and Finnish stimuli, especially that all speech items used in this study (Finnish and English) were matched in length (all were 401 ms) and in consonant and vowel duration (C-V transition point was always at 120±1 ms) which make all the items phonologically equal (for more details see supplementary table B1 and the spectrograms Figure 1). Therefore, the term CVV was corrected in the manuscript into CV as recommended by the reviewer.
Reviewer comment 4:
No problems with the analytical description or methods.
Reviewer comment 5:
My main comment for the results is the appropriateness of discussing location (i.e., mastoids) for EEG. My only experience with EEG and topo maps was to not discuss location because signals from a different part of the brain can be picked up by an electrode "far" away. Again, an expert in EEG would be able to provide more detailed feedback on this tendency.
Answer to comment 5:
We agree with the reviewer if they meant that the electrical brain activity is not equivalent to a source reconstruction, however, in the current interpretation, we did not discuss the location as a source activity but described it as a commonly used indicator in identifying the ERP responses. The electrical brain distribution obtained with HD-EEG at the scalp level and at specific latencies are ERP indicators. A description of the electrical distribution is always used in the literature to identify the ERP component such as MMR and LDN responses. For example, the electrical distribution of the MMN response is well known to have frontal negativity and a polarity reversal at the mastoids.
Reviewer comment 6:
I appreciate the detailed discussion about how the format information and co-articulation of /sh/ may have affected brain responses.
Reviewer comment 7:
Overall, the authors do a good job of placing their results within the context of the broader literature. However, the discussion would still benefit from editing to improve clarity and readability. The clarity of the discussion could be improved with more informative subheaders, key points placed at the end of paragraphs, and more paragraph breaks to space the text out.
Answer to comment 7:
Following the reviewer’s recommendations, the discussion section was edited. Subheaders and summary sentences were inserted in the revised manuscript to improve the clarity and readability of this section.
Reviewer comment 8:
There is no discussion of the strengths/limitations of the work in the discussion. Eg. the small (n = 26) sample size of children with RD. Not being able to account for differences in English exposure beyond school (do some children get more practice than others). I encourage the authors to address at a minimum the small sample size limitation and how their work can still more us forward.
Answer to comment 8:
As recommended by the reviewer, we have now added the strengths and limitations of the study in the discussion and conclusion sections.
The following paragraph addresses with some clarification the limitations and strengths of the study:
- The RD group size could cause low statistical power for some effects, and we mention in the manuscript the difference in the group size between the two groups. An additional limitation is the exposure to English outside the school which is, however, not possible to control. Exposure is expected to occur at home via TV for example, or via the Internet, and these were mentioned in the method section for clarity. However, as we are looking at the effect on the group level, there is no real need to control for it as the groups' average exposure is expected to be similar. This is still not of critical value to this research as we do not run individual analyses or comparisons where individual exposure may affect the results.
- A strength of the study is the use of two speech stimuli from different languages, rarely used in previous studies of dyslexia. Further, the same stimuli were used earlier in an independent sample of a different age group. This yields continuity to the research field investigating foreign language processing in individuals with reading difficulties.
Reviewer comment 9:
The introduction and discussion could be improved by mirroring the topics in both sections. Section headers in the introduction would help readers glean the main themes more easily (speech perception and reading; native vs foreign processing; EEG work on speech perception).
I think the addition of a section on "complications with stimuli in speech perception" in the introduction will better prime the reader for the authors discussion on /sh/, sub-phonemic information, and lexical information. The material is all there in the introduction; it just needs to be organized and tightened up a bit more.
Answer to comment 9:
As pointed out by the reviewer and to improve clarity, paragraph headers were inserted in the introduction.
The matter was addressed, and a short paragraph was added to introduce the sound /sh/ as a foreign sound in Finnish phonology.
Reviewer comment 10:
Beyond EEG responses to stimuli and some group differences, I am not sure how these data point to how speech perception skills affect measureable behavior. Please either discuss this limitation in the discussion or perform some sort of that links EEG responses to behavior. Perhaps collapsing across the groups (because the ANOVA showed no group differences) and performing a regression using amplitude (or extractable statistic) to predict reading scores.
Answer to comment 10:
Although our analysis doesn´t point directly to measurable behavior as no direct correlations were conducted in this article, the grouping was based on behavioral evaluation as presented in the reading score. Details of these behavioral evaluations are available in Table 1. It is a common practice in EEG/ERP research to study brain responses in and between different groups based on behavioral evaluations. A direct correlation between the brain responses and the reading score was previously conducted using the same groups´ data in our previous article. This was added in the manuscript as a reference and evidence that the brain ERP responses are correlated to the reading score and reading skills. We refer to our previously published paper using the same data [see ref 147].
Reviewer 2 Report
The relationship between reading skills and neuronal representation of foreign language is a meaningful topic. The authors used a passive auditory oddball paradigm and ERPs to study the relationship by comparing two groups of school-year children with distinct reading abilities in both native and foreign language contexts. The findings are quite interesting in terms of two ERP components: MMR and LDN. Generally, the paper is well-organized and nice to read. However, the reviewer suggests there should be some revisions and clarifications before being accepted for publication. Questions and comments are presented as follows for the authors to consider.
Q1. The authors are expected to clarify the validity of of comparisons of brain processes including substractions of waves for MMRs between various conditions across two blocks of experiments since the stimuli of two languages are different or not well-controlled in terms of lexicality and frequence. for example, shoe/shy/she are all real English words but sii is a psudoword; she is more familiar than shoe or shy.
Q2. Oddball paradigm is classic for eliciting MMN(mismatche negativity) or P3 while your modified version (two-blocked, two-deviant) generated MMR(both negativity and positivity). And Euclidean distance computation seems innovative here. More discussions are needed for readers to understand or replicate the design.
Q3. The reviewer is surprised at the incomplete reference list and unable to find some key references in the paper including Ylinen et al., 2019 and Yu 2019.
Q4. In Discussion Within group results, Native and foreign language processes in poor readers (Line771-784), the comparison between RD and CTR groups seems unrelevant and confusing.
Q5. The numbers of participants were not balaced between RD group(26)and CTR (86).Please clarify the reason.
Q6. Line 737-738 “The LDN response to the native sai-suu contrast was more prominent than that to the sii-suu contrast in the native language processing. The response to the shy-shoe contrast was also larger than that to the she-shoe contrast.” The two statements are of double standard. To some extent, the contrast sai-suu is more like she-shoe since Sai and She are more frequent than others. More interpretations are needed.
Suggestions.
1. Terms or units should be consistent . MMN in Table 2, 3 is MMR? S or seconds?
2. Line 125-126 are repeated in Line 187?
3. Line 405-406. One citation could be added to explain why eight electrods were used.
4. Line 625 seems to contradict with Line 609-610?
5. Line 724-725 seems not convincing.
6. The referece list is too long and tables or figures need to be concised.

Author Response
Responses to the review report 2.
The relationship between reading skills and neuronal representation of foreign language is a meaningful topic. The authors used a passive auditory oddball paradigm and ERPs to study the relationship by comparing two groups of school-year children with distinct reading abilities in both native and foreign language contexts. The findings are quite interesting in terms of two ERP components: MMR and LDN. Generally, the paper is well-organized and nice to read. However, the reviewer suggests there should be some revisions and clarifications before being accepted for publication. Questions and comments are presented as follows for the authors to consider.
Question1. The authors are expected to clarify the validity of comparisons of brain processes including substractions of waves for MMRs between various conditions across two blocks of experiments since the stimuli of two languages are different or not well-controlled in terms of lexicality and frequence. for example, shoe/shy/she are all real English words but sii is a pseudoword; she is more familiar than shoe or shy.
Answer to Question 1:
We thank the reviewer for this comment. We have added a further description of the stimuli in the methods to clarify the selection and to guide the readers to the original study by Ylinen et al. (2019) for results concerning word frequency.
The stimuli were the same as in Ylinen et al. (2019), where the frequency of use was addressed. ‘She’ is very frequently used and certainly familiar to the children, whereas ‘shy’ is less frequently used. However, the Finnish stimuli were matched with the English ones by their phonology rather than frequency, because phonology was considered more important for the comparison across languages. Native Finnish words were expected to be quite familiar to the children anyway (of course, the pseudoword could not have a lexical representation). Unfortunately, all aspects could not be matched at the same time between the Finnish and English stimuli due to the limited selection of monosyllabic words in Finnish. Since it is well-established that the MMN is very sensitive to acoustic changes, the comparison between MMNs to Finnish and English items would have been difficult or impossible, if the stimuli had been matched according to the word frequency rather than the phonetic structure.
In general, the subtraction of waves for the MMRs between the conditions is the typical analysis to conduct in ERP research in order to reveal any differences. In the current manuscript, the analysis was based primarily on the phonological processing of the stimuli, which was carefully controlled. Other possible processes and aspects of the stimuli that may be embedded within the ERP responses were further discussed in the discussion section.
Question 2. Oddball paradigm is classic for eliciting MMN (mismatch negativity) or P3 while your modified version (two-blocked, two-deviant) generated MMR (both negativity and positivity). And Euclidean distance computation seems innovative here. More discussions are needed for readers to understand or replicate the design.
Answer to question 2:
The reviewer raised concerns about possible replication. However, we think that all the details related to the design are already described in the method section and the necessary references are also mentioned. This experiment itself is a replication of the design published by Ylinen et al., 2019. The Euclidean distance was only used as a verification tool and as support for our discussion. Furthermore, if there are any specific questions/details not mentioned, we remain available for contact if someone would like to reproduce the paradigm, experiment, or analysis.
Question3. The reviewer is surprised at the incomplete reference list and unable to find some key references in the paper including Ylinen et al., 2019 and Yu 2019.
Answer to question 3:
We thank the reviewer for pointing out this error. There was a technical issue, and one full page of references went missing. The reference list was fully revised, and all missing references were reinserted in the revised manuscript.
Question 4. In Discussion Within-group results, Native and foreign language processes in poor readers (Line771-784), the comparison between RD and CTR groups seems irrelevant and confusing.
Answer to question 4:
Following the reviewer's comment, these sentences were revised and corrected/edited for clarity.
Question 5. The numbers of participants were not balanced between RD group(26) and CTR (86).Please clarify the reason.
Answer to question 5:
The reviewer raised the question of imbalanced group sizes. This study is part of a larger study examining digital reading skills. In the larger study, a large sample with no learning difficulties was necessary and here we utilized all participants available for the sub-study. Further, one reason for the unbalanced numbers is the real proportion of these subgroups in the school environment. Thus, we wanted to obtain the best possible S/N ratio for both groups by using the full data set.
Question 6. Line 737-738 “The LDN response to the native sai-suu contrast was more prominent than that to the sii-suu contrast in the native language processing. The response to the shy-shoe contrast was also larger than that to the she-shoe contrast.” The two statements are of double standard. To some extent, the contrast sai-suu is more like she-shoe since Sai and She are more frequent than others. More interpretations are needed.
Answer to question 6:
In these sentences, we are discussing the ERP/brain responses based on phonological processing and primarily on that. This was not related to the word frequency. The possible effects of word frequency and semantics on the MMR/LDN responses were later addressed and discussed in the discussion section.
Suggestions.
- Terms or units should be consistent. MMN in Table 2, 3 is MMR? S or seconds?
S1 answer: Term MMN corrected to MMR in Tables 2,3 and 4 and in figure 2.
- Line 125-126 are repeated in Line 187?
S2 answer: These sentences were revised but we did not find the repetition
- Line 405-406. One citation could be added to explain why eight electrodes were used.
S3 answer: We did not use any exact previous electrode selection, but we chose our own selection in the area of interest based on previous studies. However, for additional support, and to justify the selection, references were added in the electrode description indicating that these general electrode locations are representative of the responses of interest (see line 380).
- Line 625 seems to contradict with Line 609-610?
S4 answer: We would like to clarify to the reviewer that the description in L 609 is related to the language difference, whereas the description in L 625 is related to group difference (we did not find statistically significant difference between the two groups). The information is not contradictory; they are two different comparisons with two different findings. Furthermore, these sentences were mentioned in two separate paragraphs.
The first sentence is in paragraph 4.1. Native vs. foreign language processing: “These results showed that the brain responses to foreign language items were different than the native ones within both groups. This difference may reflect the instability of the cortical representations in foreign (English) language speech sounds processing, which remains relatively novel items compared with the native sounds even after long exposure.”
The second sentence is in paragraph 4.2. Group comparison CTR vs. RD: “Our results did not show any statistically significant differences between the groups in any of the components.”
- Line 724-725 seems not convincing.
S5 answer: Following the reviewer's comment, the sentence was revised and corrected for clarity.
- The referece list is too long and tables or figures need to be concised.
S6 answer: To answer this concern, we have already tried to make the tables and figures as concise as possible. We would be happy to get additional suggestions on how to make them more concise.
The reference list is indeed covering many studies due to the topic encompassing both EEG and behavioral studies as well as native and foreign language processing in typical readers and individuals with reading difficulties. The reference list was revised, and all unnecessary references were removed.
Round 2
Reviewer 1 Report
Thank you for your revisions. You have addressed the concerns from my review. I did notice a few small things.
1. Please be consistent acronyms. You use MMN, MMN/R, and MMR at different points in the paper for the same idea. Pick 1 and use it throughout.
2. Page 6, line 274 change "consonant-vowel-vowel" to "consonant-vowel"
3. Please be consistent about italics/no-italics when discussing stimuli. Ex. page 11 line 452 "sai-suu" no italics; whereas page 12, line 501 "sai-suu" italics.
4. Separate out your limitations/strengths from the general discussion. You begin discussing limitations on page 20, line 794, within subsection 4.3.2.2. The text from line 794 to line 812 is its own subsection.
Author Response
Please find enclosed the responses to the review report 1 – round 2
- Please be consistent acronyms. You use MMN, MMN/R, and MMR at different points in the paper for the same idea. Pick 1 and use it throughout.
Answer 1: The term MMR was used to describe both negative and positive mismatch responses throughout the manuscript. The term is now corrected in the revised manuscript.
- Page 6, line 274 change "consonant-vowel-vowel" to "consonant-vowel"
Answer 2: The term consonant-vowel was corrected in the revised manuscript.
- Please be consistent about italics/no-italics when discussing stimuli. Ex. page 11 line 452 "sai-suu" no italics; whereas page 12, line 501 "sai-suu" italics.
Answer 3: All the terms were rechecked and corrected for consistency.
- Separate out your limitations/strengths from the general discussion. You begin discussing limitations on page 20, line 794, within subsection 4.3.2.2. The text from line 794 to line 812 is its own subsection.
Answer 4: The subheading “4.4. Strengths and limitations” is now added to the subsection.
Author Response
Please find enclosed the responses to the review report 2 – round 2
Q1. Concerning the results. The authors are expected to clarify the validity of of comparisons of brain processes including substractions of waves for MMRs between various conditions across two blocks of experiments since the stimuli of two languages are different or not well-controlled in terms of lexicality and frequence.
The reviewer agree with the authors that the MMN is very sensitive to acoustic changes, but the direct comparison between MMNs to Finnish and English items remains not convincing. The autors maintain that they used the same stimuli and paradigm with Ylinen et al. (2019) but the latter only discussed between-group difference rather than between-language difference. The frequencey effect could be neglected in Ylinen et al. (2019) because they only compared the group difference, while the current study compared two different language types, in which frequency may play an effect hard to neglect. It seems that the authors need more evidence or support except that of Ylinen et al. (2019).
Answer to question 1:
To answer the reviewer´s question, we think that even the relatively infrequent words remain sufficiently “familiar” to the participants in both languages and that the memory traces are sufficiently present for all the stimuli used in our study including those in the foreign language (maybe except for the pseudoword “sii” that may have different memory trace as it could be a part of other words).
There is little evidence in the literature of the word frequency modulatory effect on the MMN response. However, according to the recent study by Jacobsen et al., 2021 who addressed this specific question on word frequency/familiarity's effect on the mismatch response, there is no significant effect of word frequency on the mismatch response.
Thus, we do not clearly see any reason for not comparing the responses between languages.
Reference:
- Jacobsen, T., Bäß, P., Roye, A., Winkler, I., Schröger, E., & Horváth, J. (2021). Word class and word frequency in the MMN looking glass. Brain and Language, 218, 104964.
For better clarity, this information was added in the discussion part (please see lines: 809 - 813 in the revised manuscript)
The meaning/lexicality of the word effect was also addressed in the discussion (please see lines: 744-750 in the revised manuscript).
Q2. Concerning the results: Oddball paradigm is classic for eliciting MMN (mismatche negativity) or P3 while your modified version (two-blocked, two-deviant) generated MMR(both negativity and positivity).
The reviewer partly agree to the authors’s response that the analysis was based primarily on the phonological processing of the stimuli, which was carefully controlled. MMN (mismatche negativity) is actually visible in the subtraction of the response to a frequently repeated standard stimulus from the response to a deviant stimulus. It is in nature a difference wave, which is often discussed or compared in one block where there is only one standard stimulus (baseline). If occasionally in different blocks, the baselines need to be the same or comparable. The reviewer is interested in how the authors could directly compare MMNs between the foreign language stimuli and the native langauge stimuli since there were two different standard stimuli with different consonants in spite of the same vowel. Does that mean different baselines? Ylinen et al. (2019) didn’t report such comparisons as they stated that “Only main effects or interactions involving the factor Group are reported, because any conclusions about dyslexia are based on differences between the groups, whereas the other factors are not per se relevant for the research questions”. Therefore, the reviewer would like more clarifications about the validity of the comparsions in RESULTS 3.1.
Answer to question 2:
To address this concern, we compared the baselines of the two standards' responses English (shoe) vs Finnish (suu) (between -100ms to 0ms) in both groups (CTR and RD) and we did not find any statistical differences between the two baselines of the standards. This justifies the comparisons between the two blocks as the baselines between the two languages were statistically similar.
Q3. Line 154: What are the cases where negative peak changed into positive and what causes the upturning. Please brief the reason for the reverse of the polarity or why the authers used MMR in place of MMN in the Introduction.
Answer to question 3:
In line 154 we refer to previous studies reporting mismatch responses in infants and young children showing a positive mismatch response (P-MMR or MMR), often described in the literature as reflecting an immature response of the mismatch negativity (e.g. Maurer et al., 2003). It was also described as reflecting the development of the automaticity of mismatch detection (e.g. Lee et al., 2012). The MMR polarity has been shown to differ between early to mid-childhood (e.g. Liu et al., 2014). Thus, we used the term MMR as it is frequently present in the literature investigating the change detection/ mismatch response in children. This MMN response is typically reported negative in adults (e.g. Ruhnau et al., 2010).
References:
- Maurer, U., Bucher, K., Brem, S., & Brandeis, D. (2003). Development of the automatic mismatch response: from frontal positivity in kindergarten children to the mismatch negativity. Clinical Neurophysiology, 114(5), 808-817.
- Lee, C. Y., Yen, H. L., Yeh, P. W., Lin, W. H., Cheng, Y. Y., Tzeng, Y. L., & Wu, H. C. (2012). Mismatch responses to lexical tone, initial consonant, and vowel in Mandarin-speaking preschoolers. Neuropsychologia, 50(14), 3228-3239.
- Liu, H. M., Chen, Y., & Tsao, F. M. (2014). Developmental changes in mismatch responses to Mandarin consonants and lexical tones from early to middle childhood. PloS one, 9(4), e95587.
- Ruhnau, P., Wetzel, N., Widmann, A., & Schröger, E. (2010). The modulation of auditory novelty processing by working memory load in school age children and adults: a combined behavioral and event-related potential study. Bmc Neuroscience, 11(1), 1-14.
Following the review question, this information was clarified in the manuscript as follows: L153 “The MMN/R is typically elicited in the oddball paradigm and expressed as a negative peak in adults, or as a positive or negative peak in infants and children, between 130 ms and 250 ms “
Q4. L657-661 Discussion 4.3.1: MMR showed a very small negative cluster over the central area for the condition she-shoe, and a positive and very brief cluster for the shy-shoe condition. Here the authors attributed the atypical postivity to the presence of the foreign onset sound /ʃ/ (pronounced /sh/) and the coarticulation effect caused by the difference between /s/ and /ʃ/. The reviewer feel confused about this interpretation because she, shy and she shared the same /ʃ/ in the block which was seperated from the other block where /s/ was involved. The positivity seemed to have little to do with different consonants but have something to do with different vowels in she and shy.
Answer to question 4:
We agree with the reviewer that the interpretation of these effects is quite complex in natural speech items because there is always a confounding effect of the coarticulation on the previous consonant, but also there is an effect of the consonant on the following vowel (as discussed in the manuscript). We think that the observed positivity, or tendency to positivity, could be related to the nativeness of the /sh/ sound (/ʃ/) that typically does not have as strong representation as the native /s/. Please note that this was an interpretation of the positivity observed only in the English block within both groups (in CTR group see line 679 - 681, and in RD group see lines 761-763 in the manuscript).
However, vowels are also contributing to this response in CV items. Therefore, the following sentence describes the coarticulation effect: “Line 662: In addition, atypical responses could be attributed to the coarticulation effect.” But also, the initial sound affects the following vowel, as described in the manuscript: “Line: 665: Formant transitions are important perceptual cues in speech processing. Their shapes vary according to the neighbor consonant affecting the identification of the following vowel “.
Q5. Line 323 “the preceding vowel” or “subsequent vowel”?
Answer to question 5:
The term “subsequent” was corrected in the manuscript.
Q6. In Table 2: MMN is MMR?
Answer to question 6:
The term MMR in Table 2 was corrected in the manuscript.